# Low Cycle Fatigue Life Evaluation of Notched Specimens Considering Strain Gradient

**DOI:** 10.3390/ma13041001

**Published:** 2020-02-23

**Authors:** Shenghuan Qin, Zaiyin Xiong, Yingsong Ma, Keshi Zhang

**Affiliations:** 1College of Civil Engineering and Architecture; Key Lab of Disaster Prevent and Structural Safety; Guangxi Key Lab Disaster Prevent and Engineering Safety; Guangxi University, Nanning 530200, China; qqsshh55@163.com (S.Q.); xiongzaiyin@163.com (Z.X.); mys@gxust.edu.cn (Y.M.); 2School of Mechanical and Traffic Engineering, Guangxi University of Science and Technology, Liuzhou 545006, China; 3College of Civil and Architectural Engineering, Nanning University, Nanning 530200, China

**Keywords:** strain gradient effect, Chaboche model, model parameters calibration, notched specimen, low cycle fatigue

## Abstract

An improved model based on the Chaboche constitutive model is proposed for cyclic plastic behavior of metal and low cycle fatigue of notched specimens under cyclic loading, considering the effect of strain gradient on nonlinear kinematic hardening and hysteresis behavior. The new model is imported into the user material subroutine (UMAT) of the finite element computing software ABAQUS, and the strain gradient parameters required for model calculation are obtained by calling the user element subroutine (UEL). The effectiveness of the new model is tested by the torsion test of thin copper wire. Furthermore, the calibration method of strain gradient influence parameters of constitutive model is discussed by taking the notch specimen of Q235 steel as an example. The hysteresis behavior, strain distribution and fatigue failure of notched specimens under cyclic loading were simulated and analyzed with the new model. The results prove the rationality of the new model.

## 1. Introduction

Due to the requirements of functional design, weight reduction, and component connection, many engineering load bearing components usually have notches or holes. Due to the stress concentration, they tend to be the weak points of the structure, resulting in fatigue cracks under cyclic loads. Therefore, it is necessary to study how the notch affects the fatigue performance of components.

It has been thought that the fatigue life of components is the same if the local stress or strain is the same, whether there is a notch or not [1,2,3]. For example, in the early nominal stress and strain method, it is believed that the fatigue life of a material is the same if its stress concentration coefficient and stress spectrum are the same. Later, the local stress and strain method [4,5] calculated the local stress-strain history of the notch by the Neuber method or finite element method, and then combined with the stress-life curve or strain-life curve of the material to predict the fatigue life. A large number of test results proved that the predicted result of this method was too conservative [6,7,8]. A large number of tests show that there is a great difference in the fatigue life duration between the notched specimen and the smooth specimen. The fatigue life of notched specimen is estimated according to ‘equal stress or strain, equal fatigue life’, and the prediction is often too low [9,10]. There are different understandings of this phenomenon. Some scholars believe that only considering the stress-strain level at the danger point of the notch cannot accurately reflect the fatigue strength of the specimen, because the strain gradient exists near the notch of the specimen, which may have an important influence on the initiation of fatigue crack [11,12,13]. Due to the existence of the gradient, the influence of the stress-strain state in a certain range around the danger point should also be considered, so the critical distance method is proposed [14,15,16,17]. The critical distance is used to determine the effective stress or strain at the hot spot localized at critical distance (point method) or by averaging stress and strain distribution over critical distance (line method). This latter method is used when taking into account the stress or strain gradient. The key of the critical distance method is to determine the critical distance, but different scholars give different definitions of the critical distance [18,19,20], which is not convenient for engineering components. Considering the existence of the stress field gradient, Yao proposed the stress field strength method in 1997 [21]. This method obtains the stress field strength by weighting the stress in the local damage area of the notch. When the stress field strength of the notch root is the same as that of smooth specimen, the fatigue life of both specimens is considered to be the same. In addition, if the effect of strain gradient on the constitutive behavior of materials is considered, the evaluation of fatigue of notch specimens will also change. Xiong et al. believe that the strain gradient causes strain hardening of material near notch, which increases the initial yield strength of the material in the high strain gradient region, which will change the local strain of the notch [8]. They found that the fatigue life assessment of notched specimens according to this mechanism was more reasonable. Many scholars have studied the effect of strain gradient on the constitutive behavior of materials. Based on J_2_ flow theory, Fleck and Hutchinson proposed the couple stress theory considering the influence of rotation gradient, and successfully explained the size effect problems such as torsion of thin copper wire and bending of thin beam [22]. Then they improved the couple stress theory and proposed the stretch and rotation strain gradient plasticity theory (SG) which can consider both tension and torsion gradient [23]. Gao and Huang (1999, 2000) proposed a strain gradient plasticity theory based on mesoscopic dislocation mechanism, namely MSG theory, which combined macroscopic plasticity theory and dislocation mechanism by establishing a multi-scale and hierarchical theoretical framework [24,25]. These types of theoretical formulas introduce high-order stress, and the equilibrium equations and boundary conditions are relatively complicated. It will be more difficult to apply them to analyze actual materials. Aifantis proposed a low-order strain gradient theory without high-order stress and additional boundary conditions [26]. The primary and secondary Laplace operators of equivalent strain are introduced into the classical plastic constitutive model to consider the effect of strain gradient on the plastic flow law of materials. The model can be used to explain the shear band and local instability in metals. Applying the low-order strain gradient theory, the strain gradient can enter the constitutive relationship by modifying the plastic hardening modulus, which is convenient to be combined with the finite element method, such as the Taylor-based nonlocal theory (TNT) proposed by Gao [27] and the conventional theory of mechanism-based strain gradient (CMSG) proposed by Huang [28]. Based on this, Qu et al. combined CMSG theory with the finite element method to study the indentation size effect in the spherical indentation test, and the predicted indentation hardness was relatively consistent with the test results [29]. Yan et al. established a two-dimensional multi particle finite element model based on TNT theory, which can better reflect the size effect of particles [30].

Previous studies of strain gradients introduced into plastic constitutive models have focused on problems under monotonic loading, with little consideration given to the common material hysteresis behavior and fatigue failure problems under alternating loads. In view of these problems, the following studies were carried out in this paper. The effect of strain gradient on cyclic plastic behavior of materials was introduced into the Chaboche constitutive model, and the torsion of fine copper wire was analyzed with the new model to verify the validity and rationality of the model. Based on this, taking the Q235 steel notched specimen as an example, the calibration method of the strain gradient influencing parameters in the constitutive model is discussed, and then the model is used to analyze the strain distribution at notch root of a notched specimen under cyclic loading and to evaluate the fatigue life of the specimen.

## 2. Materials, Specimens and Experiments

### 2.1. Smooth and Notched Specimens

The experimental materials are ordinary carbon-structured hot-rolled steel plates, grade Q235B. The mechanical properties and main chemical components of the materials are shown in Table 1 and Table 2.

There are two types of specimens: smooth specimens and notched specimens. The notch radii (R) of the notched specimens are 0.5, 1, 2, and 4 mm, and the geometric dimensions are shown in Figure 1 and Figure 2. The notch was cut by molybdenum wire and polished to ensure smooth surface. The machining accuracy is IT8-IT7, and the surface roughness is Ra25-Ra1.6.

### 2.2. Fatigue Experiment of Smooth Specimens

The MTS809 fatigue testing machine (MTS Systems Corporation, Eden Prairie, MN, USA) was used in the experiment, and the loading waveform was a sinusoidal wave with a maximum frequency of 2 Hz. Tensile and compressive fatigue experiments at constant axial strain amplitudes of 0.004, 0.005, 0.006, and 0.008 were conducted by controlling the strain in the extensometer section of the specimen. The stress-strain hysteretic curves of each experiment are shown in Figure 3, and the experimental results are shown in Table 3.

### 2.3. Fatigue Experiment of Notched Specimens

The experimental loading waveform is still sinusoidal wave, and the maximum experimental frequency is 2 Hz. For the convenience of analysis, the cyclic hysteresis behaviors of four notched specimens were simulated by the Chaboche model without considering the effect of strain gradient. When the strain amplitudes of the notch root obtained from the simulation were 0.005, 0.006, and 0.008, respectively, the strain amplitudes of the extensometer section were taken as the control strain amplitudes in the fatigue experiments of the notch specimens (Table 4). The measured nominal stress-strain hysteresis curves are shown in Figure 4 (the nominal stress and strain here are engineering stress and strain), and the experimental results are shown in Table 4.

The stress and strain in the experimental section of the smooth specimen are in a uniform state, and its cross-sectional strain is known, while the strain at the notch root of the notched specimen (crack initiation) needs to be calculated. Including or excluding the effect of strain gradient in the constitutive model will affect the calculation results of the strain at the notch root of the notched specimens, thus affecting the evaluation of the fatigue life law of the notched specimens.

## 3. Constitutive Model

### 3.1. Chaboche Constitutive Model Excluding the Effect of the Strain Gradient

When the effect of strain gradient is not considered, the yield criterion of the Chaboche constitutive model [31] is defined as:(1)$$f=\sqrt{\frac{3}{2}\left(s^{\prime }-\alpha^{\prime }\right):\left(s^{\prime }-\alpha^{\prime }\right)}-R=0,$$
where f is the yield surface function, is the deviator of stress tensor, is the deviator of the back stress tensor α, andis the radius of yield surface.

The Chaboche model divides the hardening caused by plastic deformation into two parts: isotropic hardening and kinematic hardening. Isotropic hardening is characterized by swelling of the yield surface, that is:(2)$$R=\sigma_{0}+Q\left[1-\exp\left(-bp\right)\right],$$
whereis the initial yield stress, andare material parameters determined by experiments, andis the cumulative equivalent plastic strain as follow:(3)$$p=\int \dot{p}dt,\;\dot{p}=\sqrt{\frac{2}{3}{\dot{\varepsilon }}^{p}:{\dot{\varepsilon }}^{p}},$$
where $\varepsilon^{p}$ is the plastic strain tensor, $\dot{p}$ and ${\dot{\varepsilon }}^{p}$ are their rates, respectively.

While kinematic hardening is represented by the movement of the yield surface, which is expressed by nonlinear kinematic hardening, and its expression is:(4)$${\dot{\alpha }}^{\left(k\right)}=\frac{2}{3}C^{\left(k\right)}{\dot{\varepsilon }}^{p}-\gamma^{\left(k\right)}\alpha^{\left(k\right)}\dot{p},$$
(5)$$\alpha =\sum_{k=1}^{M}\alpha^{\left(k\right)},$$
wheredenotes theth component of the back stress, andis the number of back stress components. The more back stress components, the closer the simulated stress-strain curve is to the experimental curve. Some studies have shown that a better simulation result can be obtained when is set as 2 [32], so this paper takes the same value. andare the parameters describing the characteristics of the kinematic hardening of the material. The above parameters can be calibrated by combining the measured hysteresis curve of the smooth specimen (Figure 3) with the calculation of the finite element model. The calibration results are shown in Table 5.

Substituting the parameters in Table 5, the cyclic hysteresis behavior of the smooth specimen described by this model under different strain amplitudes is in good agreement with the measured experimental results, as shown in Figure 5.

### 3.2. Chaboche Constitutive Model Including the Effect of the Strain Gradient

The effect of strain gradient on the hysteresis behavior of materials can be reflected in the isotropic hardening and kinematic hardening during the cycling process. As a preliminary analysis, we only consider the effect of strain gradient on nonlinear kinematic hardening in the model. Considering this effect, the strain gradient influence factorsandare introduced. Equation (4) of Chaboche model is rewritten as:(6)$${\dot{\alpha }}^{\left(k\right)}=K_{1}^{\left(k\right)}C^{\left(k\right)}\frac{1}{R}\left(s-\alpha \right)\dot{p}-K_{2}^{\left(k\right)}\gamma^{\left(k\right)}\alpha^{\left(k\right)}\dot{p},$$
where the strain gradient influence factorsandare defined as:(7)$$K_{1}^{\left(k\right)}=1+\beta^{\left(k\right)}\cdot {\left(1+\frac{l_{0}}{l}\right)}^{n}\cdot {\left(l_{0}\cdot \eta_{eq}\right)}^{0.5},$$
(8)$$K_{2}^{\left(k\right)}=1+\frac{\kappa^{\left(k\right)}}{{\left(1+\frac{l_{0}}{l}\right)}^{n}}\cdot {\left(l_{0}\cdot \eta_{eq}\right)}^{0.5},$$
where, $l_{0}$ is the intrinsic length of the material, which is generally considered to be related to the microstructure of the material. According to previous studies by other scholars, the intrinsic length of copper and Q235 is 0.0037 mm [23] and 0.0021 mm [33], respectively, in this paper. is the characteristic length of the specimen; when there is no notch, it is the radius or the length of the short side of the specimen; when there is a notch, it is the minimum radius of curvature of the notch root. is the size effect index, determined by the experiment. andare the strain gradient constants related to the material. For the sake of simplicity, we will not consider the correction of the strain gradients of $C^{\left(2\right)}$ andtemporarily, and take bothandas 1. Therefore, only the parametersandneed to be determined. is the equivalent strain gradient. Referring to the theory of Aifantis [26] and Tsagrakis [34], to avoid high-order stresses and additional boundary conditions, the derivative of the Mises equivalent strain to spatial coordinates is taken as the strain gradient. While:(9)$$\varepsilon_{ij}=\frac{1}{2}\left(u_{i,j}+u_{j,i}\right),$$
(10)$$\bar{\varepsilon }=\sqrt{\frac{2}{3}\varepsilon_{ij}:\varepsilon_{ij}}.$$
So, the strain gradient is:(11)$$\eta_{i}=\frac{\partial \bar{\varepsilon }}{\partial x_{i}}.$$
The equivalent strain gradient is:(12)$$\eta_{eq}=\sqrt{\eta_{1}{}^{2}+\eta_{2}{}^{2}+\eta_{3}{}^{2}}.$$

In finite element calculation, taking the eight-node hexahedron element (C3D8) as an example, the cumulative displacement $u_{c}^{n}$ and displacement increment $\Delta u^{n}$ at node *n* of the element are known. By using the shape function $N_{n}$, the incremental step displacement u of any position in the element can be obtained:(13)$$u=\overset{8}{\underset{n=1}{\sum }}N_{n}(u_{c}^{n}+\Delta u^{n}).$$
The derivative of displacement u with respect to the coordinate axis yields $u_{i,j}$, namely:(14)$$u_{i,j}=\frac{\partial u_{i}}{\partial x_{j}}=\overset{8}{\underset{n=1}{\sum }}\frac{\partial N_{n}}{\partial x_{j}}u_{i}^{n}.$$
Therefore, the following formula can be used to calculate the strain gradient in the finite element calculation:(15)$$\eta_{i}=\frac{\partial {\bar{\varepsilon }}_{n}}{\partial x_{i}}=\overset{8}{\underset{n=1}{\sum }}\frac{\partial N_{n}}{\partial x_{i}}{\bar{\varepsilon }}_{n}.$$

It should be noted that ABAQUS program (Dassault Systemes, Vélizy-Villacoublay, France) cannot calculate the strain gradient of the current incremental step using only the UMAT subroutine. In order to solve this problem, this paper uses a combination of UEL and UMAT subroutines, using the same two sets of nodes and units. The keyword ‘*EQUATION’ is used to bind the degrees of freedom between the corresponding nodes of the two sets of models to ensure that the corresponding node displacement and element deformation are consistent. Only the strain gradient is calculated when the UEL subroutine is called, and then the UMAT subroutine is called to calculate the constitutive relation. This has two advantages: (1) it is convenient for post-processing analysis; (2) the UEL subroutine for calculating the strain gradient can be combined with other UMAT subroutines to introduce the strain gradient into other constitutive models without rewriting the UEL subroutine, so as to expand the application scope. It must be pointed out that in this paper, we try to use a simplified method to characterize the inhomogeneity of strain distribution near the notch root of the notched specimen. As a preliminary study, the mechanical behavior of non-Cauchy higher-order continuum is not considered.

### 3.3. Validation of the New Model

Many experimental studies have shown that when the characteristic length of the material is close to the order of microns, the material will show a strong size effect [35,36,37], among which the classic experiment is the tensile and torsion experiment of fine copper wire conducted by Fleck et al. [23]. In the tensile test, the material did not show a significant size effect, but in the torsion test, when the diameter of the fine copper wire decreased from 170 μm to 12 μm, the normalized torque increased by two times. This phenomenon cannot be explained in the traditional continuum mechanics, and the strain gradient theory is considered as one of the effective methods to explain the size effect, which has been verified by many scholars [26,38,39].

In order to test the rationality of Chaboche model considering strain gradient, the numerical simulation of Fleck’s torsion experiment of thin copper wire is carried out. In the torsion experiment of thin copper wire conducted by Fleck, the torsional mechanical response of 170 μm copper wire is close to the response at the macro scale. Therefore, in this paper, according to the torsion curve of 170 μm copper wire given by Fleck, the model material parameters are obtained as shown in Table 6. Through the trial adjustment of the remaining torsion curves, the strain gradient material constant was obtained as shown in Table 7. The simulation results are shown in Figure 6, indicating that the model can well reflect the size effect.

## 4. Fatigue Life Evaluations of Notched Specimens

The mesh size will affect the simulation results of stress and strain at notch root and the determination of strain gradient parameters. Some scholars found that it was more appropriate to set the mesh size as 0.2 mm when studying crack propagation [40] and nonlocal damage problems [41]. However, the strain gradient at the notch is not as large as that at the crack tip. Therefore, considering the accuracy and computing time, the mesh at the notch root is refined with a size of 0.25 mm. In order to improve the calculation efficiency, only the extensometer section of the notched specimens was taken for calculation (Figure 7).

### 4.1. Material Constants Determination of the Strain Gradient Model

To determine the strain gradient at the root of the actual notched specimen, the method shown in Figure 8 is adopted in this paper: Horizontal lines were drawn on the surface of the specimen where the strain was to be measured, and high-resolution photographs of the horizontal lines were taken with an optical microscope. This photo was imported into MATLAB (MathWorks, Natick, MA, USA), and the command “rgb2gray” was used to gray it, so as to obtain a matrix storing the gray scale value of each pixel of the horizontal line. From this gray scale value matrix, it can be seen that the marked position has a gray scale value of 0 and the other position is 255. Therefore, according to the number of pixels in the width direction of the horizontal line, the width changes before and after tensile deformation at different positions of the horizontal line can be determined. Thus, the distribution of tensile strain of the horizontal line can be determined.

Specific steps to determine the material constants of the strain gradient model are as follows:Two horizontal lines were drawn at the notch root of the notched specimen for marking, and high-resolution photos of the marks were taken after magnification of 30×, 50×, 100×, and 150×, respectively, under an optical microscope. At a magnification of 30×, the whole marker line can be captured (Figure 9a), while at a magnification of more than 30×, only partial photos can be captured, so the left and right photos are taken, respectively (Figure 10).The specimen was subjected to tensile test, loaded to the nominal strain of 0.0088, and then unloaded.Under the same shooting conditions, the marker lines was photographed with the method of step 1 to obtain the photos after the tensile test (Figure 9b).

MATLAB software is used to gray the photos with the command “rgb2gray” (Figure 11). The number of pixels marked is counted and the strain distribution at the notch root of the specimen is calculated.

Theoretically, the smaller the statistical range and the width of the horizontal line, the closer the calculated result is to the true strain. However, in the case of constant magnification of the shooting, the smaller the width of the horizontal line means the fewer pixels in the width direction of the horizontal line, the lower the calculation accuracy. Therefore, taking into account the need for calculation accuracy and to be close to the real strain, combined with the consideration of the marking position and other reasons, this paper counted the pixels from the lower edge of the upper marker line to the lower edge of the lower marker line and calculated the strain, as shown in Figure 12. Thus, at 30× and 150× magnification, approximately 86 and 430 pixels can be counted, and the calculation accuracy of strain is 0.0116 and 0.0023, respectively.

Figure 13 is the strain distribution curve of the notch root of the specimen calculated from the photos taken at 30×, 50×, 100×, and 150× magnification, and the abscissa is the distance from the left to right of the notch root. We know that the width of the horizontal line will stretch in the loading direction and shrink in the length of the horizontal line after tensile test, so there may be a few pixels deviation in the photo after the tensile test (Figure 14). In addition to some inevitable errors, the results of the obtained strain distribution fluctuate, but the width of the horizontal line can be determined.

The whole image of the marker lines can be photographed at 30× magnification, but due to insufficient magnification, the number of pixels of the marker line is small, and the error of the strain calculated at the notch root will be relatively large (Figure 13a). At a magnification of more than 30×, only the part of the marker line can be photographed, so it is inevitable that the processing result will be affected by human factors if images are to be stitched together. Therefore, the strain distribution calculated at various magnification factors was superimposed together (Figure 15). On the one hand, the results obtained at different magnification can be verified by each other; on the other hand, more accurate results could be obtained as far as possible without interference from human factors.

The above tensile test was simulated using the new model proposed in this paper, which considered the effect of strain gradient, to calibrate the strain gradient constants of Q235 (see Table 8) according to the measured strain distribution in Figure 15. By using the constants in Table 8, the finite element calculation results can be obtained that are in good agreement with the measured strain distribution (Figure 16).

### 4.2. Numerical Simulation of Cyclic Hysteresis Behavior of Specimens

The strain gradient term was introduced into the Chaboche model to simulate the fatigue experiments of the Q235 notched specimens, and the effect of strain gradient could be reflected by the strain gradient influence factors. Now, a row of elements on the notch root surface of the notch specimen with a notch radius of 0.5 mm is numbered, and it is listed as 1 to 12 from the edge of the notch to the inside. The relation of the peak values of the strain gradient influence factor $K_{1}{}^{\left(1\right)}$ of these 12 elements with the cycle index is shown in Figure 17. As can be seen from the figure, the strain gradient mainly affects 3 to 4 elements at the edge of the notch, and the $K_{1}{}^{\left(1\right)}$ at the inner element is close to 1.

The cyclic hysteresis behavior of the Q235 notched specimen simulated by Chaboche model considering the effect of strain gradient is shown in Figure 18. In the figure, the measured nominal stress-strain hysteresis curves of the specimens with a notch radius of 2 mm are compared with the calculated hysteresis curves considering the effect of strain gradient, which basically coincide. The same is true for other notched specimens, which are not listed due to space limitations.

### 4.3. Fatigue Life Evaluation of Notched Specimens

The Coffin-Manson formula is used to fit the fatigue experiment results of the smooth specimens in Table 3 (Figure 19). The Coffin-Manson formula is as follows:(16)$$\frac{\Delta \varepsilon_{t}^{p}}{2}=\varepsilon_{f}{\left(2N_{f}\right)}^{c}$$
where $\Delta \varepsilon_{t}^{p}$ is the residual plastic strain range (in the case of complex stress, $\Delta \varepsilon_{t}^{p}$ is taken as $\Delta \varepsilon_{eq}^{p}$ at the edge of the notch root, and $\Delta \varepsilon_{eq}^{p}$ is the residual equivalent plastic strain range), $\varepsilon_{f}$ is the fatigue ductility coefficient, c is the fatigue ductility index, the value range is generally between −0.5 and −0.7, $N_{f}$ is the fatigue life. $\varepsilon_{f}$ and c obtained by fitting are 1.02 and −0.612, respectively.

Table 9 shows the residual equivalent plastic strain amplitude at the notch root simulated by the Chaboche model including and excluding the strain gradient effect. It should be noted that the simulation conducted to determine the experimentally controlled strain amplitude of the four notched specimens ensured that the simulated strain amplitudes at the notch root of all the four notch specimens were 0.005, 0.006, and 0.008, respectively. So, the simulated residual equivalent plastic strain amplitudes at notch root of all the four notch specimens are the same at the corresponding amplitude (Table 9, first column).

By combining the data in Table 9 and Table 4, the relationship between residual equivalent plastic strain amplitude and fatigue life of the notched specimen can be obtained, which is compared with the smooth specimen, as shown in Figure 20 and Figure 21.

As a comparison, the critical distance methods was used to analyze the experimental and simulation results. Here we choose the line method and the critical distance (L) of Q235 is 0.458 mm [42]. The average residual plastic strain amplitude within 2L is taken as the effective strain amplitude $\Delta \varepsilon_{eff}$, that is:(17)$$\Delta \varepsilon_{eff}=\frac{1}{2L}\int_{0}^{2L}\Delta \varepsilon_{eq}^{p}dx,$$
the integral path is shown in Figure 22, and the results is shown in Table 10.

Also, by combining Table 10 and Table 4, the relationship between effective strain amplitude and fatigue life of the notched specimen can be obtained, as shown in Figure 23.

The residual equivalent plastic strain amplitudes at the notch root calculated by the three models in Table 9 and Table 10 are substituted into Equation (16) to compare with the results of experiments (Figure 24).

As can be seen from Figure 24, if the effect of strain gradient is not taken into account, the estimated result deviates greatly from the measured result, and the smaller the notch radius (the larger the strain gradient), the greater the deviation. The results obtained by the critical distance method are in good agreement with the measured results when the notch radius is large. But, when the notch radius is 0.5 mm, some of the predicted results exceed the double factor region and are biased to danger. By the Chaboche model, including the effect of strain gradient, the estimated life is obviously close to the measured result, and all the predicted points are within the double factor range. This result proves that it is reasonable to consider the effect of strain gradient in the constitutive relation for the fatigue life analysis of notched specimens.

## 5. Conclusions

Strain gradient term is introduced to modify the plastic hardening modulus to improve the traditional Chaboche model which can reflect the size effect of materials.By observing the change of the marker line on the surface of the notched specimen before and after the tensile test under an optical microscope, it is confirmed that the constitutive relationship of the material is indeed affected by the strain gradient. Using this method, the model parameters can be calibrated.The Chaboche model including the effect of strain gradient can be used to analyze the strain gradient effect of materials by combining UEL and UMAT subroutines.The strain at the notch root can be directly compared with the strain–fatigue life curve of the smooth specimen to predict the fatigue life of the notched specimen.

The above results demonstrate that it is reasonable to consider the effect of strain gradient in constitutive relation for improving the fatigue life evaluation of notched specimens, but it is only preliminary. For more complex cases and considerations, the selection of model parameters and the relationship between the characteristic length and notch radius need to be further studied.

## Figures and Tables

**Figure 1 materials-13-01001-f001:**
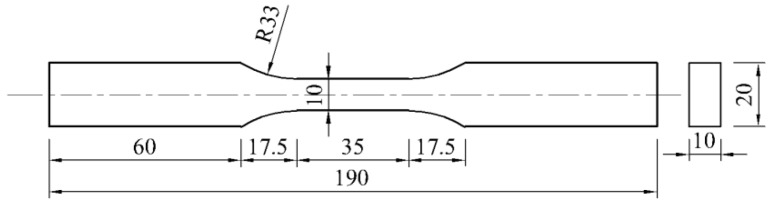
The geometric dimension of smooth specimens (unit: mm).

**Figure 2 materials-13-01001-f002:**
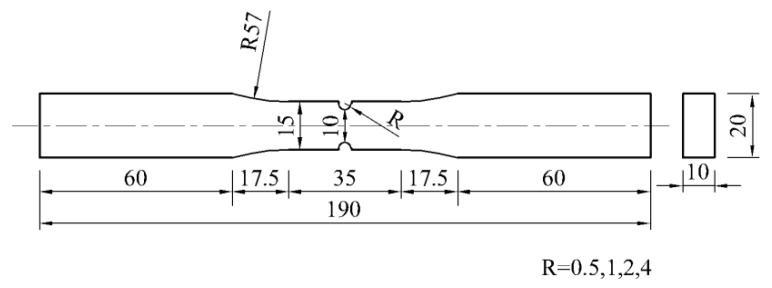
The geometric dimension of notched specimens (unit: mm).

**Figure 3 materials-13-01001-f003:**
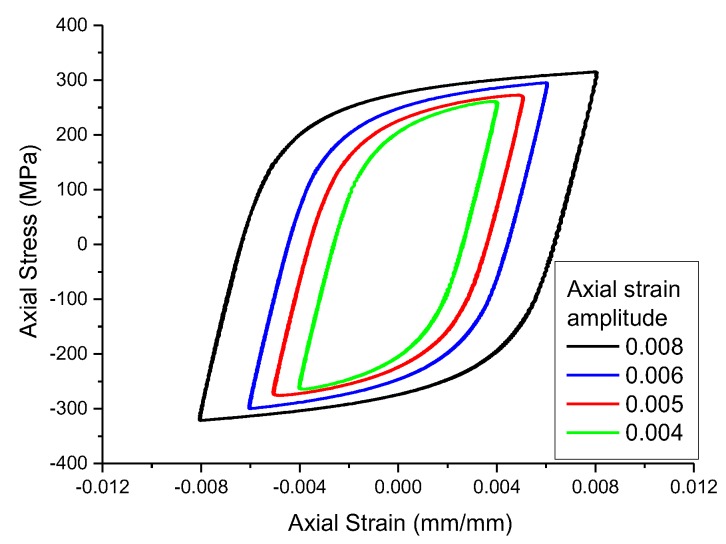
The stress-strain hysteresis curves of smooth specimens.

**Figure 4 materials-13-01001-f004:**
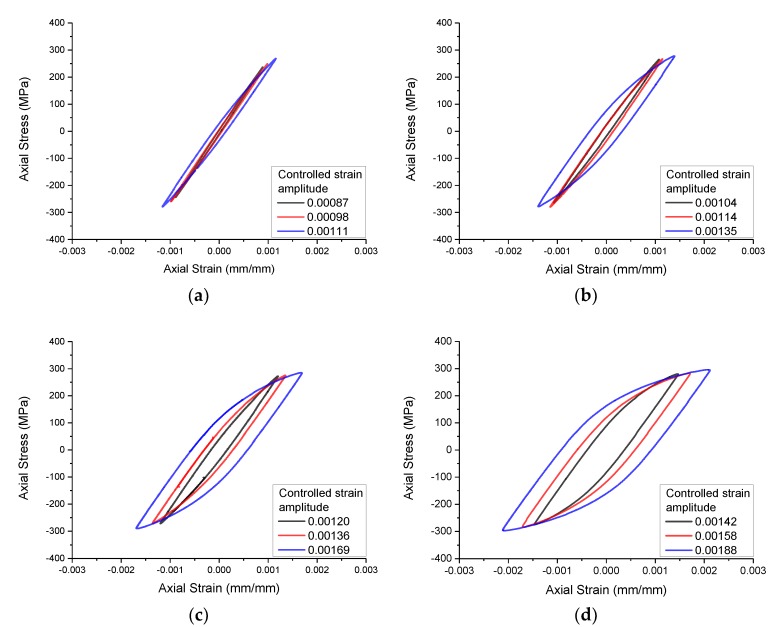
Nominal stress-strain hysteresis curves of notched specimens. (**a**) R = 0.5 mm; (**b**) R = 1.0 mm; (**c)** R = 2.0 mm; (**d**) R = 4.0 mm.

**Figure 5 materials-13-01001-f005:**
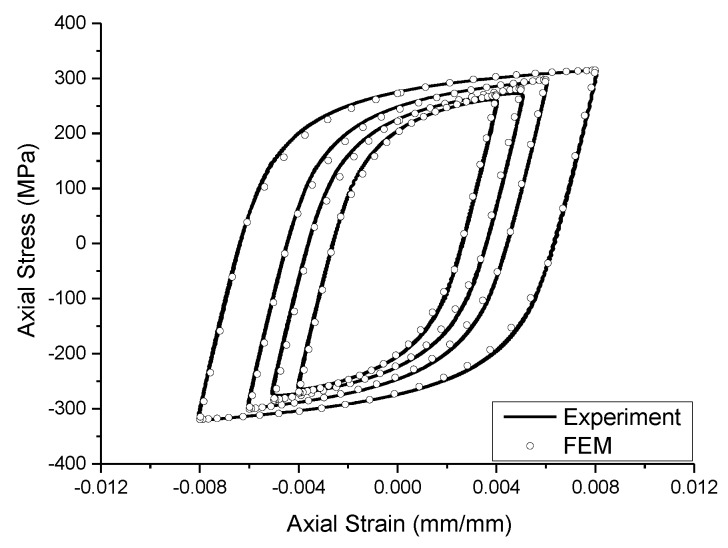
The stress—strain hysteresis curve simulated by finite element method and measured by experiment.

**Figure 6 materials-13-01001-f006:**
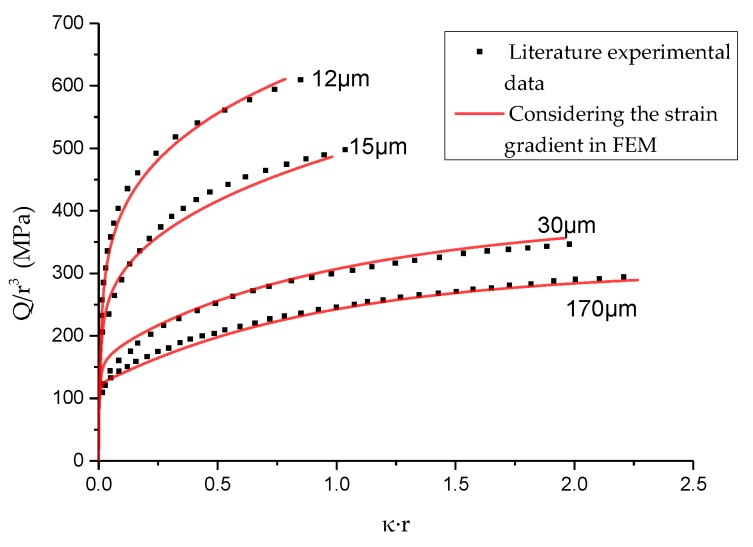
The simulation results of torsion experiment of thin copper wire.

**Figure 7 materials-13-01001-f007:**
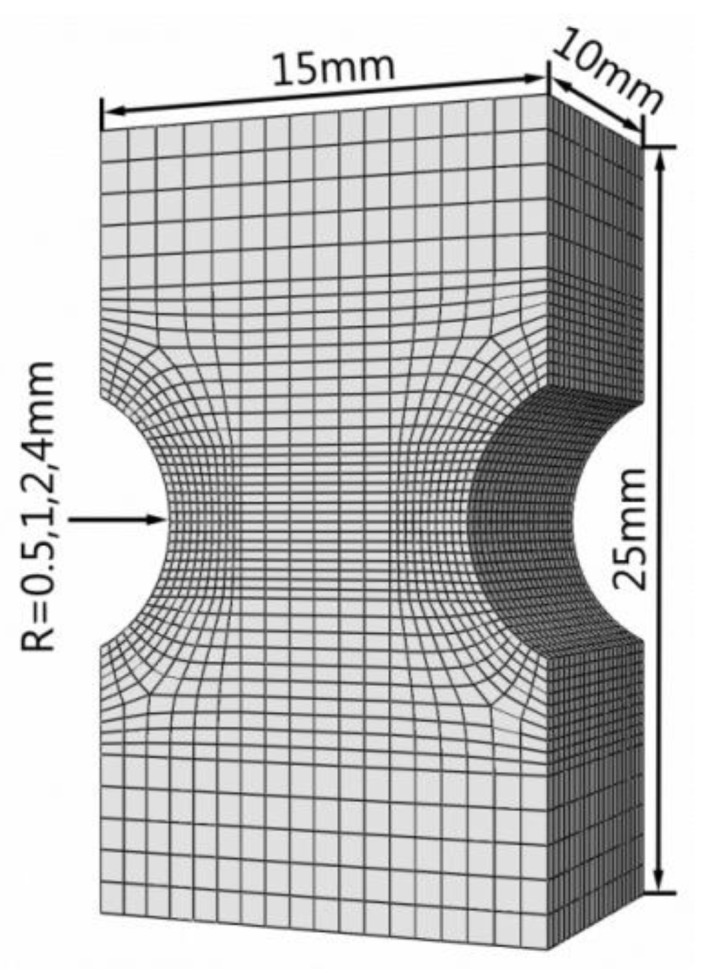
Finite element model of notched specimen.

**Figure 8 materials-13-01001-f008:**
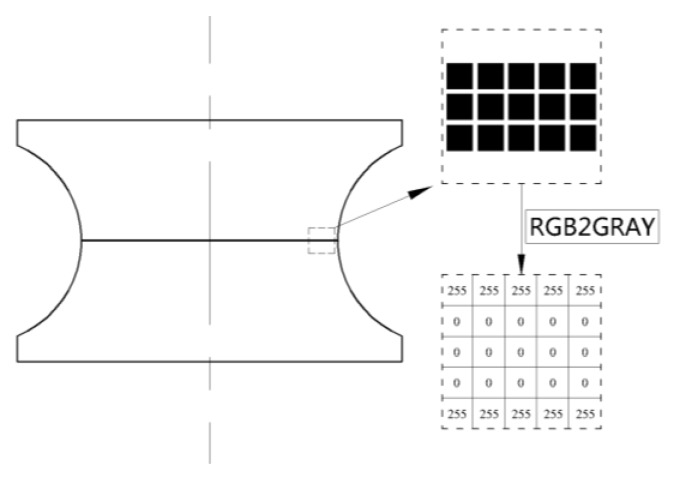
Schematic diagram of the method for measuring the strain of a specimen.

**Figure 9 materials-13-01001-f009:**
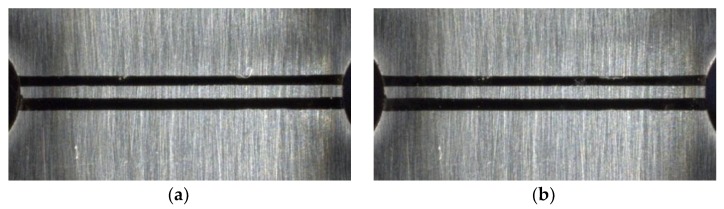
Marker lines before and after the tensile test (30×). (**a**) Before the tensile test; (**b**) After the tensile test.

**Figure 10 materials-13-01001-f010:**
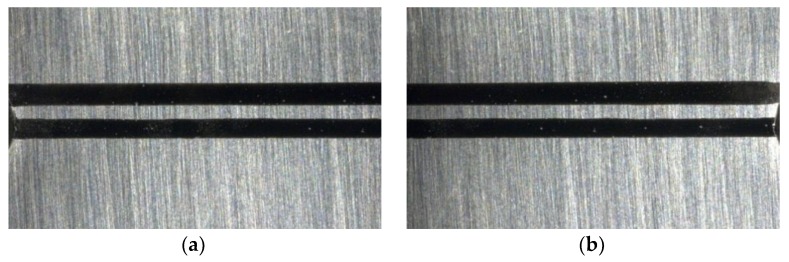
Marker lines before the tensile test (50×). (**a**) Left; (**b**) Right.

**Figure 11 materials-13-01001-f011:**
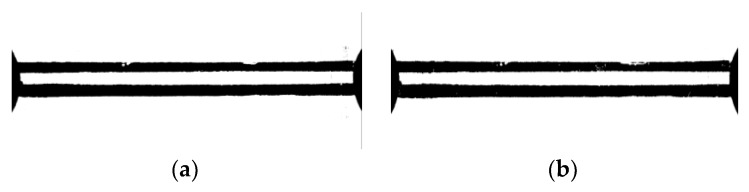
The image of the marker lines after gray processing (30×). (**a**) Before the tensile test; (**b**) After the tensile test.

**Figure 12 materials-13-01001-f012:**
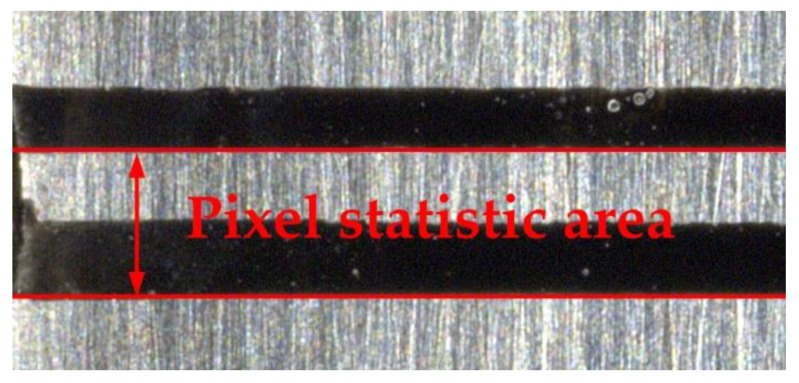
The pixel statistical area.

**Figure 13 materials-13-01001-f013:**
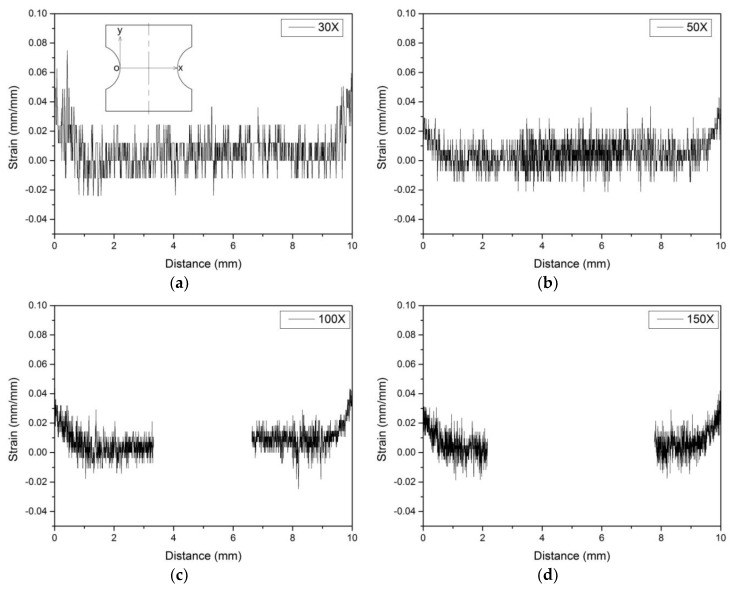
Strain distribution at the notch root. (**a**) 30×; (**b**) 50×; (**c**) 100×; (**d**) 150×.

**Figure 14 materials-13-01001-f014:**
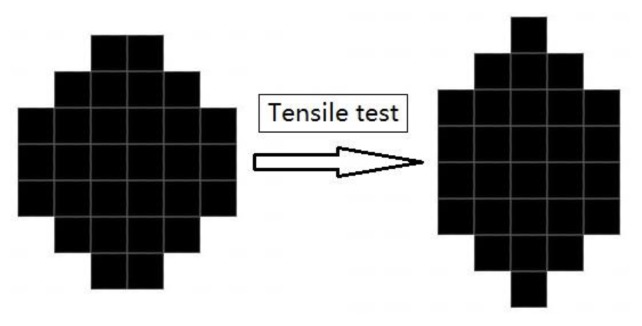
Pixel deviation diagram.

**Figure 15 materials-13-01001-f015:**
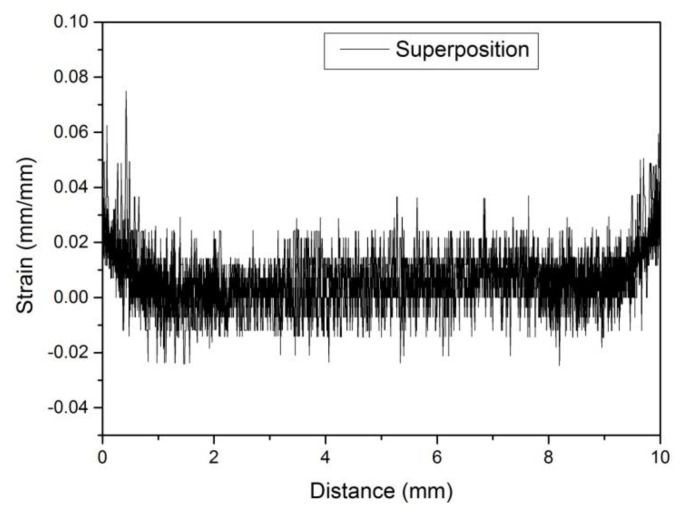
The superposition results of strain distribution at the notch root obtained from the photos with different magnification.

**Figure 16 materials-13-01001-f016:**
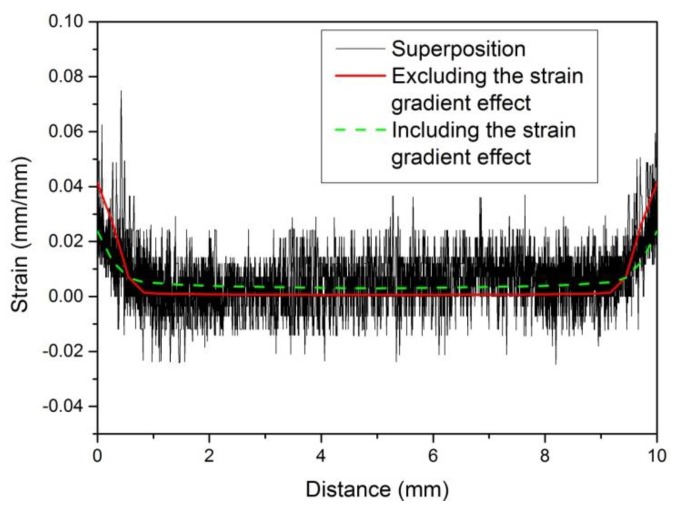
The numerical simulation results are compared with the measured results.

**Figure 17 materials-13-01001-f017:**
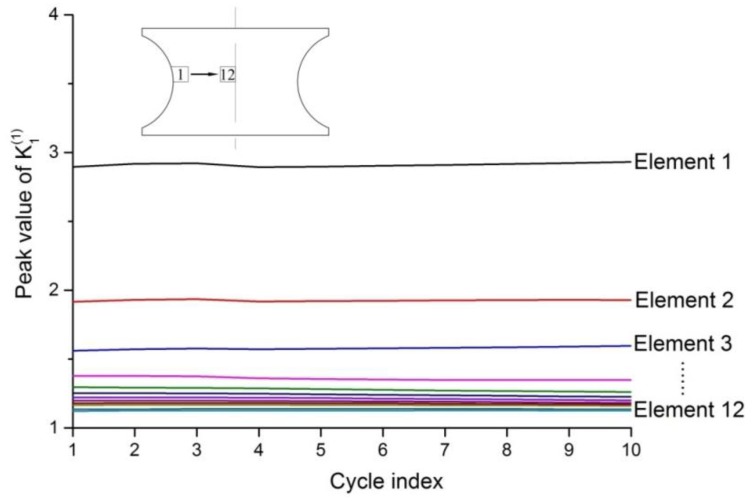
The relation of the peak values of $K_{1}{}^{\left(1\right)}$ of 12 elements on the notch root surface with the cycle index.

**Figure 18 materials-13-01001-f018:**
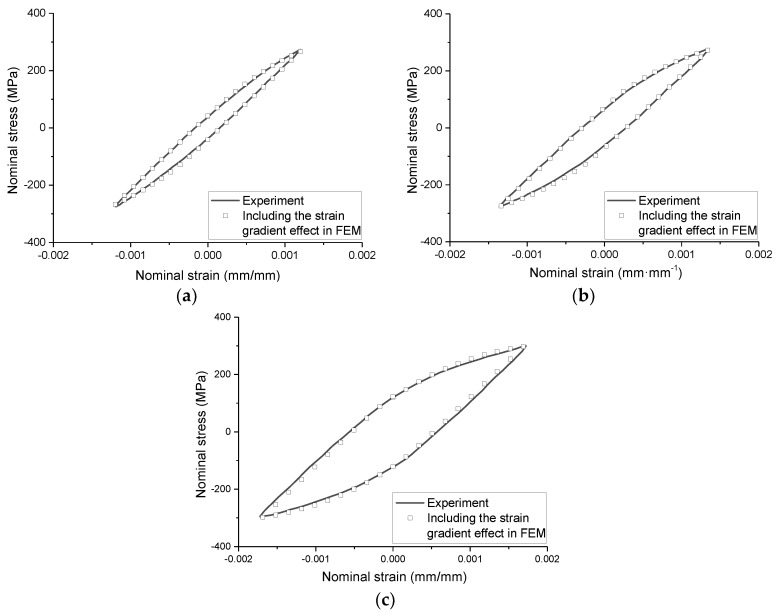
Measured and simulated nominal stress-strain curves. (**a**) Controlled strain amplitude is 0.00120; (**b**) Controlled strain amplitude is 0.00136; (**c**) Controlled strain amplitude is 0.00169.

**Figure 19 materials-13-01001-f019:**
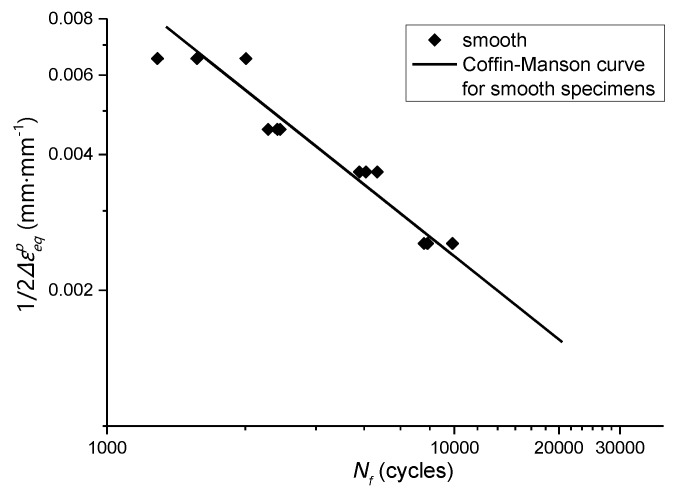
Relationship between residual equivalent plastic strain amplitude and fatigue life of smooth specimens.

**Figure 20 materials-13-01001-f020:**
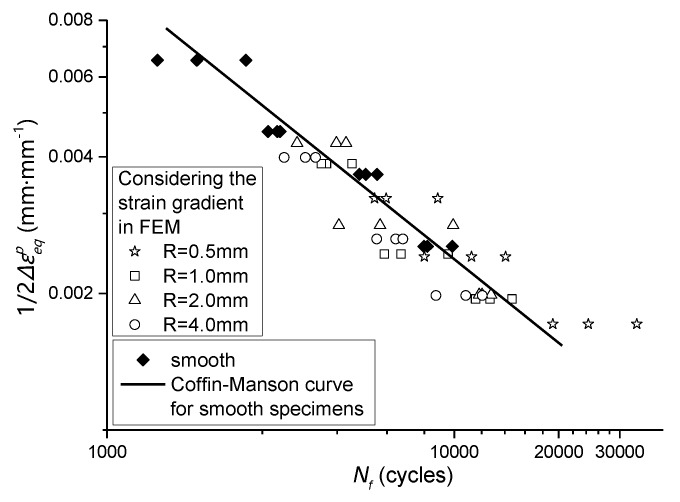
The residual equivalent plastic strain amplitude considering the effect of strain gradient -fatigue life relationship of notched specimens was compared with that of smooth specimens.

**Figure 21 materials-13-01001-f021:**
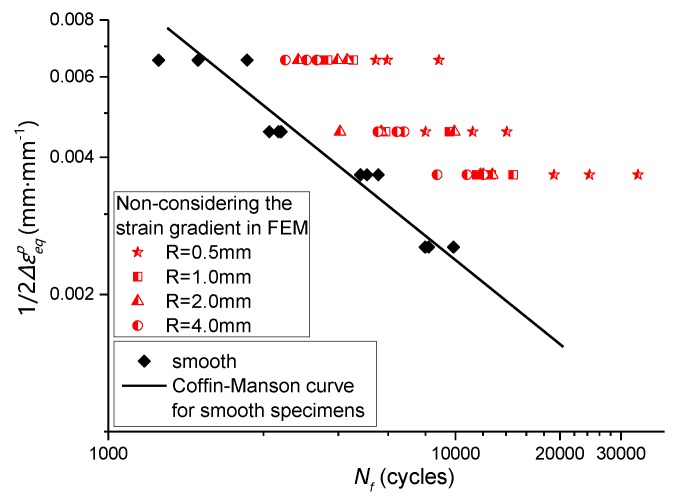
The residual equivalent plastic strain amplitude without considering the effect of strain gradient-fatigue life relationship of notched specimens was compared with that of smooth specimens.

**Figure 22 materials-13-01001-f022:**
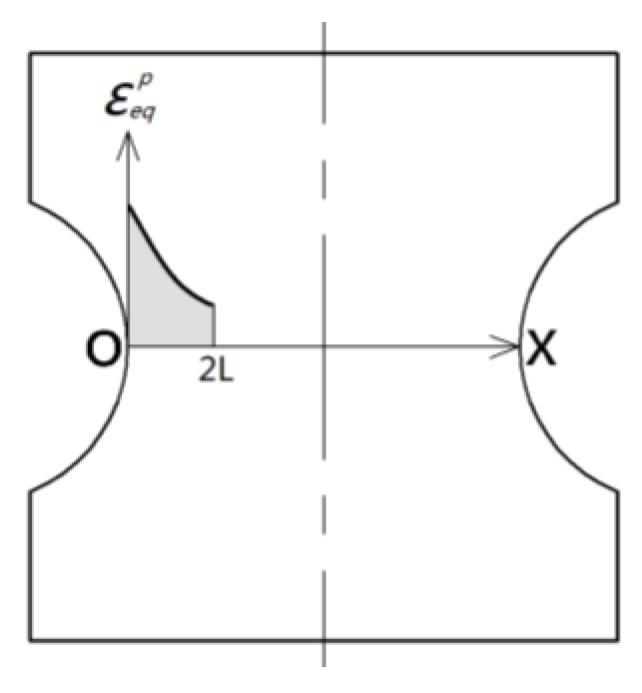
The integral path of line method.

**Figure 23 materials-13-01001-f023:**
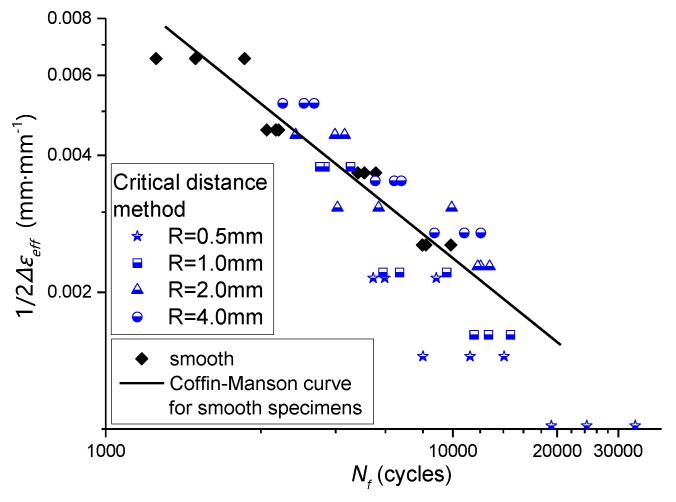
The effective strain amplitude-fatigue life relationship of notched specimens was compared with that of smooth specimens.

**Figure 24 materials-13-01001-f024:**
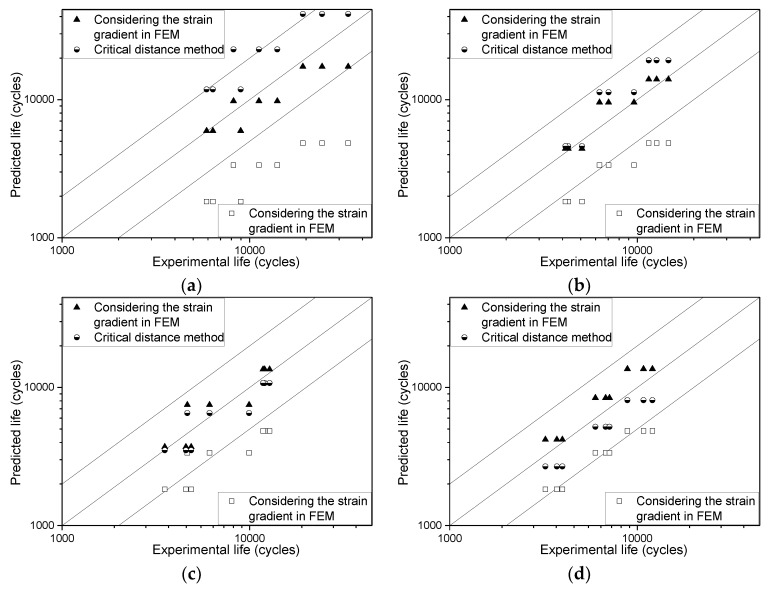
The estimated fatigue life compared with the measured results. (**a**) R = 0.5 mm; (**b**) R = 1.0 mm; (**c**) R = 2.0 mm; (**d**) R = 4.0 mm.

**Table 1 materials-13-01001-t001:** The mechanical properties of Q235B.

Yield Strength (MPa)	Tensile Strength (MPa)	Young’s Modulus (MPa)	Poisson’s Ratio	Elongation
260	550	193900	0.277	20%

**Table 2 materials-13-01001-t002:** Main chemical components of Q235B (%).

C	Si	Mn	P	S	As	Cep
0.14	0.032	0.4	0.03	0.019	0.031	0.42

**Table 3 materials-13-01001-t003:** Fatigue experimental results of smooth specimens.

Axial Strain Amplitude	Fatigue Life (Cycles)			Average Fatigue Life (Cycles)
	Specimen 1	Specimen 2	Specimen 3	
0.004	8369	8183	9877	8810
0.005	6000	5565	5332	5632
0.006	3094	2911	3151	3052
0.008	1814	1397	2512	1908

**Table 4 materials-13-01001-t004:** Fatigue experimental results of notched specimens.

Notched Specimen	Controlled Strain Amplitude	Fatigue Life (Cycles)			Average Fatigue Life (Cycles)
		Specimen 1	Specimen 2	Specimen 3	
R = 0.5 mm	0.00087	24,351	33,570	19,238	25,720
	0.00098	11,217	8210	14,059	11,162
	0.00111	5900	6370	8967	7079
R = 1.0 mm	0.00104	14,680	11,524	12,687	12,964
	0.00114	6294	7034	9611	7646
	0.00135	4287	5078	4143	4503
R = 2.0 mm	0.00120	12,781	12,032	11,800	12,204
	0.00136	6117	4654	9935	6902
	0.00169	3529	4578	4882	4330
R = 4.0 mm	0.00142	8863	12,050	10,800	10,571
	0.00158	6784	4654	9935	6628
	0.00188	3727	3239	3990	3652

**Table 5 materials-13-01001-t005:** Chaboche constitutive model parameters of Q235B.

*σ*_0_ (MPa)	*Q* (MPa)	*b*	C^(1)^ (MPa)	*γ* ^(1)^	C^(2)^ (MPa)	*γ* ^(2)^
245	−78	200	98,000	1150	9500	800

**Table 6 materials-13-01001-t006:** The material parameters of copper.

Young’s Modulus (MPa)	Poisson’s Ratio	*σ*_0_ (MPa)	*Q* (MPa)	*b*	*C*^(1)^ (MPa)	*γ* ^(1)^	*C*^(2)^ (MPa)	*γ* ^(2)^
108,500	0.326	62	158	2.5	16,000	520	2000	200

**Table 7 materials-13-01001-t007:** The strain gradient material constants of copper.

*β* ^(1)^	*κ* ^(1)^	*β* ^(2)^	*κ* ^(2)^	*n*
353.5	568.8	1.0	1.0	3.0

**Table 8 materials-13-01001-t008:** The strain gradient material constants of Q235.

*β* ^(1)^	*κ* ^(1)^	*β* ^(2)^	*κ* ^(2)^	*n*
545.0	6.5	1.0	1.0	3.0

**Table 9 materials-13-01001-t009:** Simulation results of residual equivalent plastic strain amplitude at the notch root.

Excluding the Strain Gradient in FEM	Including the Strain Gradient in FEM			
	R = 0.5 mm	R = 1.0 mm	R = 2.0 mm	R = 4.0 mm
0.00366	0.00171	0.00194	0.00199	0.00198
0.00455	0.00241	0.00244	0.00283	0.00264
0.00653	0.00324	0.00386	0.00429	0.00399

**Table 10 materials-13-01001-t010:** Effective strain amplitude calculated by line method.

R = 0.5 mm	R = 1.0 mm	R = 2.0 mm	R = 4.0 mm
0.00102	0.00161	0.00228	0.00270
0.00145	0.00221	0.00307	0.00352
0.00215	0.00377	0.00443	0.00521

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
