# Peer review of "Low Cycle Fatigue Life Evaluation of Notched Specimens Considering Strain Gradient"

_materials, 2020, doi:10.3390/ma13041001_

Round 1

Reviewer 1 Report

In the reviewer’s opinion, the present contribution cannot be accepted for publication for the several methodological and theoretical issues listed here below:

Damage mechanics can be a tool to use to describe cracking, no matter if caused by monotonic or cyclic loadings. The link between fracture and damage mechanics, and how damage models can be adopted in a standard displacement-based FE analysis (as done here with Abaqus) is never discussed. How is mesh dependence affecting the results? It is well known and reported in blogs related to it, that within a UEL you have to include everything needed for a UMAT alone. Hence, it is not clear why claiming that both were used. Looking at Fig. 3 and many to follow, it is not clear how important the effect of kinematic hardening is in comparison to the isotropic one. Apparently, it is marginal and so much of the discussion seems only a way to complicate matters with no remarkable benefits. Important: strain is a second-order tensor and hence strain gradient is a third-order tensor. What shown in Eq. 11 is the gradient of the equivalent strain, which is not the whole strain gradient. This issue is not discussed at all: it can be correct, but it must be stated and not granted for sure. People working on non-Cauchy or higher-order continua would find this inappropriate or misleading. 15 is wrong: N is the matrix used to interpolate displacements. Strain components, as space derivatives of the displacement field, is interpolated through the matrix usually called B, which contains the spatial derivatives of the shape function. How can you use the same matrix to interpolate fields that are one the derivative of the other? This is absolutely incorrect from the theoretical/mathematical point of view. Finally, reported results look good in the numerical/experimental comparison since all the parameters in the model have been tuned in order to match the same experimental data. Authors should better show the accuracy of the same tuned model under different loading conditions, in order to see if – despite the number of issues discussed here above – the method still works well.

Besides this, the grammar must be thoroughly improved throughout the whole manuscript, as it currently looks rather poor in this regard.

Author Response

Response to Reviewer 1 Comments

Firstly, we are very grateful to reviewer for the comments.

Point 1: Damage mechanics can be a tool to use to describe cracking, no matter if caused by monotonic or cyclic loadings. The link between fracture and damage mechanics, and how damage models can be adopted in a standard displacement-based FE analysis (as done here with Abaqus) is never discussed.

Response 1: The damage at the root of the notch will definitely affect the crack initiation and fatigue life, which is very important for the study of the fatigue failure mechanism. However, this paper mainly studies how to improve the fatigue life evaluation of notched specimens. The cracking description and the relationship between fracture and damage mechanics are not the main concerns of this paper.

The related statement above has been reflected in the introduction and conclusion sections.

Point 2: How is mesh dependence affecting the results?

Response 2: The mesh size will affect the simulation results of stress and strain at notch root and the determination of strain gradient parameters. Some scholars found that it was more appropriate to set the mesh size as 0.2 mm when studying crack propagation [41] and nonlocal damage problems [42]. However, the strain gradient at the notch is not as large as that at the crack tip. Therefore, considering the accuracy and computing time, in this manuscript the mesh at the notch root is refined with a size of 0.25 mm.

This explanation has been added in Page 8 of the new version.

Point 3: It is well known and reported in blogs related to it, that within a UEL you have to include everything needed for a UMAT alone. Hence, it is not clear why claiming that both were used.

Response 3: All the functions of the UMAT can be included in UEL, but the post-processing becomes more cumbersome if only the UEL subroutine is used for simulation. We use UEL and UMAT together, mainly considering the convenience of strain gradient calculation. Because UEL can deal with the strain calculation of all nodes and integration points in the element at the same time, and UMAT only calculates one integration point each time, obviously the latter is more convenient for the calculation of variable gradient. Further, it will increase the complexity of programming if all the constitutive relation calculations are carried into the UEL. For the sake of simplification and reliability, we choose a computing strategy that combines UEL and UMAT.

We have added the related statement above in Page 7.

Point 4: Looking at Fig. 3 and many to follow, it is not clear how important the effect of kinematic hardening is in comparison to the isotropic one. Apparently, it is marginal and so much of the discussion seems only a way to complicate matters with no remarkable benefits.

Response 4: For the problem of low cycle fatigue cycle, whether the constitutive model can reasonably describe the hysteresis behavior of materials is the most important. The hysteresis behavior of materials under cyclic loading can be described by the combination model of kinematic hardening and isotropic hardening. Generally speaking, kinematic hardening is more important to describe the overall hysteresis behavior. The effect of strain gradient on the mechanical behavior of materials should also be tested according to the hysteresis behavior shown in Figure 3.

Point 5: Important: strain is a second-order tensor and hence strain gradient is a third-order tensor. What shown in Eq. 11 is the gradient of the equivalent strain, which is not the whole strain gradient. This issue is not discussed at all: it can be correct, but it must be stated and not granted for sure. People working on non-Cauchy or higher-order continua would find this inappropriate or misleading.

Response 5: Yes, strain is a second-order tensor and hence strain gradient is a third-order tensor, and shown in Eq. 11 is the gradient of the equivalent strain, it is not the whole strain gradient. Here a simplified method is tried to take the gradient of equivalent strain as a representation of the non-uniformity of strain distribution near the notch of the notched specimens. As a preliminary study, the mechanical behavior of non-Cauchy and higher-order continuum is not considered.  

We have added the related statement above in Page 8.

Point 6: 15 is wrong: N is the matrix used to interpolate displacements. Strain components, as space derivatives of the displacement field, is interpolated through the matrix usually called B, which contains the spatial derivatives of the shape function. How can you use the same matrix to interpolate fields that are one the derivative of the other? This is absolutely incorrect from the theoretical/mathematical point of view.

Response 6: N is the matrix used to interpolate displacements, this is right. But by N matrix, the other variables are also able to be interpolated. “The finite element approximation is based on assuming interpolations, by which displacement, position, and—often—other variables at any material point are defined by a finite number of nodal variables.” This can be found in the “Abaqus 6.14 online documentation - ABAQUS theory guide - 1.2.1 notation”.

Point 7: Finally, reported results look good in the numerical/experimental comparison since all the parameters in the model have been tuned in order to match the same experimental data. Authors should better show the accuracy of the same tuned model under different loading conditions, in order to see if – despite the number of issues discussed here above – the method still works well.

Response 7: This is a good question. In this paper, the model is used to simulate the fatigue experiment of four notched specimens with three controlled strain amplitudes, and the predicted fatigue life is in good agreement with the experimental results. In order to prove “the accuracy of the same tuned model under different loading conditions”, much more researches need to be conducted. This is our next goal, we will study it further.

Point 8: Besides this, the grammar must be thoroughly improved throughout the whole manuscript, as it currently looks rather poor in this regard.

Response 8: Yes, this is our weakness. We have tried our best to improve it.

Reviewer 2 Report

The aim of this paper  is evaluate low cycle fatigue life of notched specimens made in steel. It is well known since Neuber that the effective strain which rules the fatigue life duration is not the hot spot stress or strain but a lower one.  Several methods have been proposed to reduce the hot spot stress or stain : to take the stress or strain at a given distance of the loading distribution  (point  method) or averaging over another distance the strain or stress distribution (line method).

 Authors have chosen another way to reduce the hot spot strain by calculating it at the hot spot with theconstitutive Chaboche’s  model (with isotropic and kinematic hardening) coupled with a strain gradient effect. This approach is semi-original because some line methods are also coupled with strain gradient and have been proposed in literature. This model seems to give interesting prediction of fatigue life but suffers of the following criticism:

 The model needs 14 materials constants. This is very heavy model with 14 constants which fit easily any experimental results .

The model suffers of the same difficulties that point and line methods  i.e  it use a characteristic length which is considered as constant and equal to notch radius. This is the weak point of all models to evaluate low cycle fatigue life of notched specimens .It is not discuss and the choice of the characteristic length equal as notch radius  is unusual. In addition, the model needs an intrinsic length related to material microstructure without any precisions.

This paper is correctly written with logical sequences. English is correct and pictures quality is good.  The main interest of this paper is to take into account for this problem the strain gradient. For this reason it is accepted but with major revisions  (discussion about the interest of this method when comparing with point and line method and more justifications for intrinsic and characteristic length.

Details

 Page 1” its stress concentration coefficient is the same as its stress spectrum.” This sentence is unclear

 “In the fatigue law” more precisely in the fatigue life duration

 Page 2 The critical distance is used to determine the effective stress or strain at the hot spot localised at critical distance (point method) or by averaging stress and strain distribution over critical distance. This later method is used when taking into account stress or strain gradient

   “local hardening of the notch”, strain hardening of material near notch

“TNT” “CMSG “define these acronyms

 Page  3

 MPa and not Mpa in Table 1

Extensiometer and not extensiometer

 What is the “nominal” stress-strain hysteretic  (hysteresis ) curves why nominal

 Page 5

“directional hardening” is generally called kinematic hardening

“Initial yield stress” initial yield stress

Page 6 “ hysteretic” hysteresis

Page 7”  when there is a notch, it is the minimum radius of curvature of the notch root” why this definition is different from the Paris’s one (notch radius divided by 2)

 Page 8  “fleck” Fleck

 Page 12 “ hysteretic” hysteresis

Page 13 “ hysteretic” hysteresis

page 14  “the simulated strain amplitudes at the notch root” i.e . the local strain amplitude is taken as notch root?

 Fig 22 (a)”R=0.5mm; (b)R=1.0mm” (a)R=0.5mm; (b)R=0.5mm

Author Response

Response to Reviewer 2 Comments

The aim of this paper is evaluate low cycle fatigue life of notched specimens made in steel. It is well known since Neuber that the effective strain which rules the fatigue life duration is not the hot spot stress or strain but a lower one. Several methods have been proposed to reduce the hot spot stress or stain: to take the stress or strain at a given distance of the loading distribution (point method) or averaging over another distance the strain or stress distribution (line method).

Authors have chosen another way to reduce the hot spot strain by calculating it at the hot spot with the constitutive Chaboche’s model (with isotropic and kinematic hardening) coupled with a strain gradient effect. This approach is semi-original because some line methods are also coupled with strain gradient and have been proposed in literature. This model seems to give interesting prediction of fatigue life but suffers of the following criticism:

 The model needs 14 materials constants. This is very heavy model with 14 constants which fit easily any experimental results.

The model suffers of the same difficulties that point and line methods i.e. it use a characteristic length which is considered as constant and equal to notch radius. This is the weak point of all models to evaluate low cycle fatigue life of notched specimens .It is not discuss and the choice of the characteristic length equal as notch radius is unusual. In addition, the model needs an intrinsic length related to material microstructure without any precisions.

This paper is correctly written with logical sequences. English is correct and pictures quality is good. The main interest of this paper is to take into account for this problem the strain gradient. For this reason it is accepted but with major revisions (discussion about the interest of this method when comparing with point and line method and more justifications for intrinsic and characteristic length.

Response: We are very grateful to reviewer for the comments, suggestions and encouragement. This study is an attempt to estimate the root strain of notched specimens by considering the cyclic plastic constitutive relation of strain gradient, so as to improve the Manson-Coffin formula to evaluate the fatigue test life of notched specimens. The results in this paper prove that it is reasonable to consider the effect of strain gradient in constitutive relation, but it is only preliminary. We agree with the reviewer that for more complex cases, the selection of model parameters and the relationship between characteristic length and notch radius need to be further studied. We have stated this in the conclusion section of the manuscript new version.

Details

Page 1

Point 1:” its stress concentration coefficient is the same as its stress spectrum.” This sentence is unclear.

Response 1: Thanks! We have changed this sentence to:”… it is believed that the fatigue life of a material is the same if its stress concentration coefficient and stress spectrum are the same.”

Point 2: “In the fatigue law” more precisely in the fatigue life duration

Response 2: We have made correction according to the reviewer’s comments.

Page 2

Point 3: The critical distance is used to determine the effective stress or strain at the hot spot localised at critical distance (point method) or by averaging stress and strain distribution over critical distance. This later method is used when taking into account stress or strain gradient.

Response 3: Thanks! We have revised this sentence on page 2 adopting the reviewer’s suggestion.

Point 4: “local hardening of the notch”, strain hardening of material near notch

Response 4: This has been corrected according to the reviewer’s comments.

Point 5: “TNT” “CMSG “define these acronyms

Response 5: We have given definitions of these abbreviations in their original positions on page 2.

SG - the stretch and rotation strain gradient plasticity theory.

TNT - Taylor-based nonlocal theory.

CMSG – the conventional theory of mechanism-based strain gradient.

Page  3

Point 6: MPa and not Mpa in Table 1

Response 6: Sorry, we are careless! These have been corrected.

Point 7: Extensiometer and not extensiometer

Response 7: Thanks! This has been corrected.

Point 8: What is the “nominal” stress-strain hysteretic (hysteresis) curves why nominal

Response 8: Here, the nominal stress is the engineering stress or Lagrange stress, and the nominal strain is the corresponding engineering strain. In order to avoid misunderstandings, we have added explanations there.

Page 5

Point 9:“directional hardening” is generally called kinematic hardening

Response 9: We have made correction according to the reviewer’s comments.

Point 10:“Initial yield stress” initial yield stress

Response 10: This has been corrected.

Page 6

Point 11: “ hysteretic” hysteresis

Response 11: Thanks! We've changed all the “hysteretic” to “hysteresis” in this manuscript.

Page 7

Point 12:” when there is a notch, it is the minimum radius of curvature of the notch root” why this definition is different from the Paris’s one (notch radius divided by 2)

Response 12: The different definition has no effect on formulas (7) and (8) due to its undetermined constant.

Page 8

Point 13: “fleck” Fleck

Response 13: Sorry, we are careless! These have been corrected.

Page 12

Point 14: “ hysteretic” hysteresis

Response 14: We have made correction.

Page 13

Point 15: “ hysteretic” hysteresis

Response 15: We have made correction.

page 14 

Point 16: “the simulated strain amplitudes at the notch root” i.e . the local strain amplitude is taken as notch root?

Response 16: Yes, it means that the local strain amplitude at the notch root of the specimens is controlled to be 0.005, 0.006 and 0.008, respectively, excluding the strain gradient in FEM. Then the residual equivalent plastic strain amplitudes at notch root can be obtained and shown in Table 9.

Point 17: Fig 22 (a)”R=0.5mm; (b)R=1.0mm” (a)R=0.5mm; (b)R=0.5mm

Response 17: We have made correction (Fig 24 in new version). Thanks!

Reviewer 3 Report

The current paper presents interesting research and combines experimental results with the calibration of a Chaboche-type model for LCF. However, I've got the impression that the paper has not been written and typesetted thoroughly. Often, spaces are missing, the typesetting of formula and text is inconsistent, which makes the paper hard to read. Furthermore, the resolution of images should be increased, and the paper should be checked very carefully for language errors. Please also see the attached pdf for further comments.

Overall, the paper seems to have scientific merit. However, it looks like the authers did not take enough time to write the paper and present the content in a reader-friendly way. Therefore, I suggest major revision.

Author Response

Response to Reviewer 3 Comments

The current paper presents interesting research and combines experimental results with the calibration of a Chaboche-type model for LCF. However, I've got the impression that the paper has not been written and typesetted thoroughly. Often, spaces are missing, the typesetting of formula and text is inconsistent, which makes the paper hard to read. Furthermore, the resolution of images should be increased, and the paper should be checked very carefully for language errors. Please also see the attached pdf for further comments.

Overall, the paper seems to have scientific merit. However, it looks like the authers did not take enough time to write the paper and present the content in a reader-friendly way. Therefore, I suggest major revision.

Response:

We are very grateful to the reviewer for his comments and suggestions. We have revised each item according to the comments in the PDF file (See the revised version uploaded, Word version).

Reviewer 4 Report

Some general comments: 

The work is very well written. It is very well structured. The results are well presented. The conclusions are in accordance with the results. 

Anyway, I think that the proposed method for measuring strain based on image analysis needs explanation in terms of pixel measurement accuracy. 

Some line by line comments:

Page 7

Line 9  “...root. n is the size effect index, determined by the experiment. k å’Œ k   are the strain”

There is a chinese character.

Page 8

Line 17  “...of thin copper wire conducted by fleck, the torsional mechanical response of 170μm copper wire is”

There is a chinese character.

Page 10

Figure 9:   I don't see any difference between the figure 9a and figure 9b.

Figure 11:   I don't see any difference between the figure 11a and figure 11b.

Page 16

Figure 22  “Figure 22. The estimated fatigue life compared with the measured results. (a)R=0.5mm; (b)R=1.0mm; (c)R=1.0mm; (d)R=1.0mm.

Points (c) and (d) have the wrong radius: (c)R=2.0mm; (d)R=4.0mm.”

Author Response

Response to Reviewer 4 Comments

Some general comments:

Point 1: The work is very well written. It is very well structured. The results are well presented. The conclusions are in accordance with the results.

Response 1: We are very grateful to the reviewer for his comments, suggestions and suggestions.

Point 2: Anyway, I think that the proposed method for measuring strain based on image analysis needs explanation in terms of pixel measurement accuracy.

Response 2: At 30 magnification, there are approximately 86 pixels per column in the pixel statistic area. At 150 magnification, it's 430 pixels. So the calculation accuracy of strain is 0.0116 and 0.0023, respectively. This statement has been added in Page 10.

Some line by line comments:

Page 7

Point 3: Line 9  “...root. n is the size effect index, determined by the experiment. k å’Œ k   are the strain”

There is a chinese character.

Response 3: We have made correction.

Page 8

Point 4: Line 17  “...of thin copper wire conducted by fleck, the torsional mechanical response of 170μm copper wire is”

There is a chinese character.

Response 4: Does it mean the word fleck is not capitalized? We have made correction for this. But we didn’t find the Chinese character in Line 17 of Page 8.

Page 10

Point 5: Figure 9:   I don't see any difference between the figure 9a and figure 9b.

Response 5: The deformation at 30x is so small that the difference may be hard to detect. Now zoom in on the notch in figure 9a and figure 9b, put them together and add auxiliary lines for easy viewing (Figure 2).

Figure 2. Partial enlargement of figure 9.

Point 6: Figure 11:  I don't see any difference between the figure 11a and figure 11b.

Response 6: Also, zoom in on the notch in Figure 11a and Figure 11b, put them together and add auxiliary lines as shown in Figure 3.

Figure 3. Partial enlargement of figure 11.

Page 16

Point 7: Figure 22  “Figure 22. The estimated fatigue life compared with the measured results. (a)R=0.5mm; (b)R=1.0mm; (c)R=1.0mm; (d)R=1.0mm.”

Points (c) and (d) have the wrong radius: (c)R=2.0mm; (d)R=4.0mm.”

Response 7: We have made correction. Thank you!

Round 2

Reviewer 1 Report

The Authors did nothing to show that they are right and I instead was wrong in my former comments. They basically stated, without changing anything critical or consistent, that the approach they followed is this one and we must be convinced it is the right one: I'm not, for all the reasons I already wrote in my first report.

Hence, my judge does not change.

Reviewer 2 Report

 The revised manuscript has been strongly improved by reconsidering the problem of critical distance. Therefore the paper is accepted.

Reviewer 3 Report

- fix typesetting errors in Eqs. (3), (4), (6), please check the pdf!
- Table 9 goes over two pages, please fix this.